# The ADCK Kinase Family: Key Regulators of Bioenergetics and Mitochondrial Function and Their Implications in Human Cancers

**DOI:** 10.3390/ijms26125783

**Published:** 2025-06-17

**Authors:** Noel Jacquet, Yunfeng Zhao

**Affiliations:** Department of Pharmacology, LSU Health Sciences Center in Shreveport, 1501 Kings Highway, Shreveport, LA 71130-3932, USA; noel.jacquet@lsuhs.edu

**Keywords:** ADCK1, ADCK2, ADCK3, ADCK4, ADCK5, human cancer, bioenergetics, mitochondria

## Abstract

AarF domain-containing kinases (ADCKs) are a family of putative mitochondrial proteins that have been implicated in various aspects of mitochondrial function and cellular metabolism. Mitochondria play a crucial role in cellular bioenergetics, primarily in adenosine triphosphate (ATP) production, while also regulating metabolism, thermogenesis, apoptosis, and reactive oxygen species (ROS) generation. Evidence suggests that the ADCK family of proteins is involved in maintaining mitochondrial architecture and homeostasis. In detail, these proteins are believed to play a role in processes such as coenzyme Q biosynthesis, energy production, and cellular metabolism. There are five known isoforms of ADCK (ADCK1–ADCK5), some of which have similar activities, and each also has its own unique biological functions. Dysregulation or mutations in specific ADCK isoforms have been linked to several pathological conditions, including multiple human cancers, primary coenzyme Q10 (CoQ10) deficiency, and metabolic disorders. This review surveys the current body of peer-reviewed research on the ADCK protein family, incorporating data from the primary literature, case studies, and experimental studies conducted in both in vitro and in vivo systems. It also draws on existing review articles and known published findings to provide a comprehensive overview of the functional roles, disease associations, and molecular mechanisms of ADCK proteins. Further in-depth research on ADCK proteins has the potential to unlock critical insights into their precise mechanisms. This could pave the way for identifying new therapeutic targets for mitochondrial and metabolic-related diseases, as well as for advancing cancer treatment strategies.

## 1. Bioenergetics and Mitochondrial Function

Cellular bioenergetics encompasses the intricate processes by which a cell generates, utilizes, and stores energy to sustain its vital functions. At its core, bioenergetics involves the production of adenosine triphosphate (ATP), the cell’s primary energy currency, through various metabolic pathways such as glycolysis, oxidative phosphorylation, and fatty acid oxidation. Energy production supports essential biological functions, including protein synthesis, cell proliferation, signal transduction, and the maintenance of ion gradients across membranes. In addition to ATP production, bioenergetics also involves the regulation of energy storage mechanisms, including the conversion of excess glucose into glycogen or fats for later use. This tightly regulated network allows cells to adapt to fluctuating energy demands and maintain metabolic homeostasis [1,2,3,4,5].

Mitochondria are central to this adaptive capacity. Mitochondria regulate ATP synthesis using both short-term and long-term mechanisms, including mitochondrial fusion and fission, as well as adjustments in ATP synthesis flux. Disruptions in mitochondrial function can interfere with the cell’s metabolic homeostasis, leading to shifts in the balance of cellular bioenergetics. Importantly, disruptions in cellular bioenergetics—especially those involving mitochondrial dysfunction—are increasingly recognized as key contributors to the pathogenesis of numerous human diseases. Impaired ATP production and altered redox balance can lead to metabolic imbalances, increased oxidative stress, and cell death [6,7,8].

Mitochondrial dysfunction is closely associated with oxidative stress as well as alterations in mitochondrial dynamics, protein synthesis, innate immunity, and overall cellular homeostasis. Such dysfunctions are implicated in a wide spectrum of disorders, including neurodegenerative diseases (e.g., Parkinson’s and Alzheimer’s), cardiovascular conditions, metabolic syndromes like diabetes, and various forms of cancer. Kinase activity plays a crucial role in regulating various mitochondrial functions, including respiration, reactive oxygen species (ROS) generation, apoptosis induction, mitochondrial dynamics (fusion/fission), and cellular signaling. Several well-studied kinase signaling cascades are specifically targeted to the mitochondria, where they modulate mitochondrial processes and their associated products. For instance, protein kinase families such as PKA-PKC, PI3K-AKT/PKB, and Raf-MEK-ERK are known to influence mitochondrial respiration rates, control mitochondrial permeability, and regulate apoptotic pathways. These kinases exert their effects by either promoting or inhibiting mitochondrial functions that are vital for maintaining cellular homeostasis. The influence of various kinase families underscores their complex role in regulating cellular stress responses, metabolic fluctuations, and responses to stimuli. Many kinase families already serve as druggable targets for the treatment of human diseases. As a result, understanding and targeting bioenergetic pathways serves as an emerging area of therapeutic interest with the potential to transform the treatment of energy-related diseases [9,10,11,12,13,14,15]. 

## 2. Aarf Domain-Containing Kinases (ADCK)

The AarF domain-containing kinase (ADCK) family of proteins is a group of putative mitochondrial kinases believed to play a pivotal role in regulating mitochondrial functions and cellular metabolism. Currently, five isoforms of ADCK proteins have been identified. Although the precise functions of each isoform remain largely undefined, the ADCK kinase family is involved in coenzyme Q (CoQ) biosynthesis, maintenance of mitochondrial homeostasis, and tumorigenesis. Several ADCK family proteins are highly conserved through evolution in *Drosophila*, *Caenorhabditis elegans* (*C. elegans*), and mammals. These kinases are thought to be involved in critical mitochondrial activities, including the regulation of oxidative phosphorylation and the biosynthesis of CoQ, a crucial component of the electron transport chain essential for ATP production. Despite growing interest in the ADCK family, the molecular mechanisms underlying the specific roles of each isoform in mitochondrial homeostasis, energy metabolism, and cellular stress responses are not fully understood [16,17,18,19,20,21,22].

Further investigation is essential to unravel the precise roles of ADCK proteins in mitochondrial biology and their broader implications in human disease. Members of the ADCK family have been identified recently for their involvement in regulating mitochondrial homeostasis, oxidative stress, and cellular signaling. However, the molecular mechanisms underlying their activity remain largely undefined. Emerging evidence also suggests that several ADCK isoforms play key roles in the pathogenesis of various cancers, likely through their influence on mitochondrial function and cellular metabolism. Understanding the precise molecular functions of the ADCK family could provide valuable insights into the regulation of cellular bioenergetics and open new avenues for therapeutic strategies targeting mitochondrial disorders [17,23,24,25,26,27,28].

The known oncogenic mechanisms influenced by ADCK family members include increased cell proliferation, dysregulated cell cycle progression, immune evasion, tumor microenvironment modulation, increased invasiveness, and resistance to anticancer therapies. These effects often depend on the expression patterns and functional specificity of individual ADCK isoforms. Notably, all of the five known human ADCK isoforms have been implicated in regulating oxidative stress—a critical factor in cancer development and progression, as reactive oxygen species (ROS) influence DNA damage, cell survival, and signaling pathways [17,23,24,25,26,27,28].

Beyond these functions, the ADCK protein family also plays a role in several additional cellular processes with direct implications for oncogenesis. These include modulation of the inflammatory response, alterations in cellular energy metabolism, regulation of kinase signaling cascades, and control of mitochondrial dynamics. Since mitochondria are central to energy production, apoptosis, and redox balance, disruptions in their function by ADCK proteins may contribute to the metabolic reprogramming seen in many cancers. In addition, the kinase activity associated with ADCK proteins suggests their potential involvement in broader signaling networks that coordinate growth and survival pathways in cancer cells. These roles position the ADCK family as important molecular regulators in tumor biology and potential therapeutic targets in oncology (Figure 1) [14,19,23,24,26,29].

The ADCK family has been implicated in a wide spectrum of human diseases. Across the family, ADCK proteins are linked to both inherited metabolic and neurodegenerative disorders, such as cerebellar ataxia and nephropathy, as well as various cancers, including lung, breast, colon, prostate, and melanoma. Their involvement in tumor biology spans key processes such as cell proliferation, apoptosis, oxidative stress response, metabolic adaptation, immune evasion, and drug resistance. These proteins contribute to both tumor-promoting and protective pathways, depending on the context, highlighting their dual roles in disease and their potential as therapeutic targets (Table 1) [30,31,32,33,34,35].

Further investigation is needed to elucidate how these kinases contribute to mitochondrial function and their potential involvement in various diseases, particularly those related to mitochondrial dysfunction. Understanding the precise molecular functions of the ADCK family could provide valuable insights into the regulation of cellular bioenergetics and open new avenues for therapeutic strategies targeting mitochondrial disorders.

## 3. ADCK1

AarF domain-containing protein kinase 1 (ADCK1) is believed to be a mitochondrial protein involved in the regulation of the maintenance of mitochondrial structures and homeostasis. The ADCK1 gene is located on the long arm of chromosome 14 at position 24.3 (14q24.3). It spans a genomic region composed of 17 exons, which are transcribed into multiple alternatively spliced mRNA variants. In yeast, overexpression of ADCK1 has been shown to compensate for growth defects observed in Mdm10 mutants. Mdm10 is a key component of the ER-mitochondria encounter structure (ERMES) complex, which plays a critical role in maintaining lipid homeostasis within mitochondria. The synthesis and transport of lipids are essential for proper mitochondrial biogenesis and function. In this context, ADCK1 appears to help regulate mitochondrial lipid levels, ensuring adequate lipid availability for mitochondrial membranes. Additionally, ADCK1 was seen to modulate the ERMES complex, highlighting its crucial role in maintaining the structural integrity and functional dynamics of mitochondria. These data underscore the importance of ADCK1 in coordinating mitochondrial lipid metabolism and supporting overall mitochondrial health [16,18,25,58,59,60].

A study of ADCK1 in *Drosophila* revealed that the loss of ADCK1 leads to developmental defects, premature death, and mitochondrial dysfunction. The deletion of ADCK1 resulted in the premature death of *Drosophila* larvae during the second instar phase, mediated by a molting defect that induced hypoxic stress and subsequent mortality. However, re-expression of ADCK1 was able to rescue the larval lethality caused by its knockout. The expression of this protein was found to be essential in various tissues for the complete viability of the animals. Mutations in ADCK1 were associated with reduced size and lifespan, indicating its crucial role in the overall health and development of *Drosophila* larvae. ADCK1 has been shown to directly impair mitochondrial function in *Drosophila*. The loss of ADCK1 leads to a reduction in both membrane potential and ATP production. This effect on cellular metabolism may be influenced by the interaction of ADCK1 with key mitochondrial proteins such as OPA1, YME1L1, and IMMT within mitochondria-specific signaling pathways. These findings highlight the necessity of ADCK1 in sustaining mitochondrial functions [18,36].

In humans, ADCK1 has been identified as playing a role in the development or regulation of several human diseases. Genetic alterations in the ADCK1 gene have been linked to cases of schizophrenia. A study examining the effects of paliperidone palmitate treatment in patients with schizophrenia (n = 159) found that the response to this treatment was influenced by polymorphisms in the ADCK1 gene. Specifically, two significant SNPs (rs11159291 and rs12590199) and one haplotype (rs2364747–rs12590199) were identified as contributing to the treatment response. The mechanisms by which the variations in the ADCK1 gene interact with paliperidone palmitate are still unknown. However, these findings point to the potential of these polymorphisms as biomarkers for predicting responses to paliperidone palmitate treatment [37,61].

A study of mutations in parathyroid cancer (PC) patients found recurrent mutations in the ADCK1 gene. Researchers found a recurring missense mutation that altered the codon produced. Mutations in this gene occurred in 11.8% of PC cases analyzed. The observed mutations were classified as having a dominant-negative effect, where the monoallelic somatic mutation of the gene interferes with the normal functions of the wild-type protein. This observation suggests that ADCK1 may serve as a candidate oncogene in the development of PC [38].

In a study examining ADCK1 expression in osteosarcoma (OS) cells, researchers found that the knockout of ADCK1 led to significant disruptions in mitochondrial function. ADCK1 expression was found to be elevated in human OS tissues, suggesting a potential role in cancer progression. The depletion of ADCK1 resulted in a marked reduction in ATP production and the activation of apoptotic pathways. Furthermore, ADCK1 knockout led to a decrease in mitochondrial membrane potential, ATP depletion, and increased production of reactive oxygen species (ROS). In contrast, the overexpression of ADCK1 was associated with enhanced cellular proliferation and migration in OS cells. In vivo studies revealed that ADCK1 knockdown or knockout inhibited the growth of pOS-1 xenografts in nude mice, further supporting its role in promoting tumorigenesis. Collectively, these findings indicate that the overexpression of ADCK1 contributes to a pro-cancerous phenotype, while its depletion or knockout exerts anti-cancer effects in OS cells. This suggests that ADCK1 may function as an oncogene, playing a critical role in the development and progression of osteosarcoma [25].

In colon cancer, ADCK1 has been associated with pro-cancerous effects through its interaction with the β-catenin/TCF signaling pathway. Research indicates that ADCK1 expression is upregulated in colon cancer patients and is negatively correlated with patient survival. In vitro studies have demonstrated that the increased expression of ADCK1 in colon cancer cell lines enhances their ability to form colonies and increases their invasive potential. Conversely, the knockdown of ADCK1 not only mitigated these effects but also promoted apoptosis. In vivo experiments have shown that knocking down ADCK1 inhibited the ability of cancer cells to form organoids and tumors, while also reducing metastasis. These effects are thought to be mediated by ADCK1’s interaction with TCF4, a transcription factor with essential functions in the Wnt/β-Catenin signaling cascade and development. ADCK1 appears to facilitate the interaction between β-Catenin and TCF4, thereby activating the pro-cancerous functions of the Wnt/β-Catenin pathway. The overexpression of ADCK1 activates the Wnt/β-Catenin reporter gene, leading to a more malignant cell phenotype. The role of ADCK1 in colon cancer underscores its significant oncogenic potential, positioning it as a promising therapeutic target for future treatment strategies. Current knowledge of ADCK1’s involvement in mitochondrial functions and tumorigenesis suggests that it is a viable research target for metabolic diseases [33,62].

## 4. ADCK2

AarF domain-containing kinase 2 (ADCK2) is a protein-coding gene located on chromosome 7 at position q34 (7q34). The gene spans approximately 22,165 base pairs and comprises eight exons. The ADCK2 protein has been identified as a mitochondrial protein, with specific localization to the mitochondrial matrix. The ADCK family of genes has been implicated in the biosynthesis of (CoQ). CoQ plays a vital role in the inner mitochondrial membrane as both an antioxidant and a cofactor in oxidative phosphorylation. Deficiencies in CoQ can lead to oxidative stress, increased cell death, and mitochondrial dysfunction. Specifically, ADCK2 has been linked to impaired mitochondrial lipid metabolism, resulting in myopathy and CoQ deficiency. This effect arises from ADCK2 haploinsufficiency, highlighting its role in energy homeostasis and mitochondrial function. ADCK2-dependent CoQ deficiency has been associated with impaired metabolic function, reduced fatty acid β-oxidation, and diminished physical performance. In humans, ADCK2 haploinsufficiency leads to decreased mitochondrial CoQ levels. In vivo, studies on ADCK2-heterozygous mice revealed significant skeletal muscle defects at birth. However, progressive CoQ supplementation appeared to mitigate these effects, improving both skeletal muscle integrity and mitochondrial function. These findings suggest that ADCK2 haploinsufficiency should be classified as a mitochondrial disorder due to its profound impact on mitochondrial functions. Furthermore, CoQ supplementation may prove to be a potential therapeutic strategy to counteract the deficiencies caused by ADCK2 insufficiency [23,39,41,42,63].

ADCK2 may play a significant role in the development and progression of various human cancers. Emerging research suggests that alterations in ADCK2 expression or function can influence tumorigenesis and metabolic regulation. In breast cancer, ADCK2 expression has been correlated with tumor size and may serve as a predictive marker for treatment response (N = 40 or 37). Markedly, elevated ADCK2 expression has been observed in luminal A breast cancer patients, with evidence indicating that estrogen receptor (ER)-positive breast cancer cells may depend on ADCK2 for survival. This dependency suggests that ADCK2 expression levels could serve as a biomarker for predicting treatment efficacy. A 2022 study reported missense mutations in ADCK2 in 9.4% of breast cancer patients, whereas such mutations were relatively rare in studies of other cancers, occurring in 0–1% of endometrial carcinomas and up to 3.5% of uterine carcinomas. Despite the low mutation frequency, the observed correlation between ADCK2 expression and tumor size underscores the need for further research to elucidate its role in tumorigenesis and its potential as a therapeutic target [39,40,43,64,65].

Knockout studies of ADCK2 in prostate and osteosarcoma cells have demonstrated a reduced impact of tumor necrosis factor-alpha (TNFα) on hypoxia-inducible factor 1-alpha (HIF-1α), suggesting a potential role for ADCK2 in hypoxia-driven tumor progression. ADCK2 is believed to mediate its effects through the RELB-dependent NF-κB signaling pathway, which plays a crucial role in inflammatory responses, cell survival, and metabolic adaptation in cancer. The RELB/NF-κB axis is known to regulate reactive oxygen species (ROS) production, and ADCK2 may influence this process by modulating superoxide generation. Since ROS production is essential for cancer cell proliferation, survival, and metastatic potential, disruptions in ADCK2 function could alter oxidative stress responses and tumorigenic signaling. In this study, ADCK2 was established as a crucial regulator of TNFα-induced HIF-1α accumulation. The application of siRNA knockdown techniques led to a notable decrease in HIF-1α levels within prostate cancer cell lines. Furthermore, the expression of ADCK2 was demonstrated to be directly modulated by TNFα in U2OS osteosarcoma cells. These findings underscore the role of ADCK2 in the regulatory pathway associated with HIF-1α in the context of cancer biology. Given the critical role of these pathways in tumor progression, further research is needed to elucidate the precise molecular mechanisms by which ADCK2 regulates NF-κB signaling, redox homeostasis, and oncogenesis. Understanding these interactions could provide insights into potential therapeutic strategies targeting ADCK2 in cancer treatment [34].

In the context of melanoma, ADCK2 emerges as a crucial regulator of cell motility, a vital component in the processes of tumorigenesis and metastasis. The ability of cancer cells to move and invade surrounding tissues significantly contributes to the progression of the disease. Modulating the expression of ADCK2 has the potential to induce notable changes in both cell viability and motility, primarily through its regulation of MYL6, a key player in cellular dynamics. The knockdown of ADCK2 led to a downregulation of TYR and TRP1, indicating potential changes in the migration and invasion phenotypes of the cells. This study also identified a positive correlation among ADCK2, RAB2A, and MYL6, implying that ADCK2 expression influences the regulation of oncogenic and cytoskeletal proteins. Additionally, the knockdown of MYL6 mitigated the effects of ADCK2 expression in melanoma cells. This interplay suggests that manipulating ADCK2 levels could offer insights into therapeutic strategies aimed at mitigating tumor spread and enhancing treatment outcomes [35].

Similarly, ADCK2 has been implicated in the regulation of oncogenic processes in non-small cell lung cancer (NSCLC). Bioinformatic analyses of NSCLC patient data revealed a negative correlation between ADCK2 expression and overall survival, along with anti-PD-1/PD-L1 therapeutic response, suggesting a potential prognostic role. In vivo studies of ADCK2 knockout led to a reduction in cellular proliferation, viability, motility, and cell cycle progression, indicating its role in tumor growth and progression. Conversely, ADCK2 overexpression promoted oncogenic phenotypes, further supporting its involvement in NSCLC pathogenesis. Mechanistically, the loss of ADCK2 appears to inactivate the Akt-mTOR signaling cascade, a critical pathway regulating cell growth, metabolism, and survival. Conversely, the overexpression of ADCK2 enhanced the activation of this pathway through the phosphorylation of Akt and S6K1. The phosphorylation of Akt and S6k1 was significantly decreased in ADCK2-KO xenografted tumors. This depleted ATP levels in the tumors and inhibited the activation of the Akt-mTOR pathway. There was also an increase in TBAR activity, which suggests an increase in oxidative stress or mitochondrial injury. This disruption results in the suppression of malignant characteristics, reinforcing the hypothesis that ADCK2 may function as an oncogene in NSCLC development. However, further studies are necessary to fully elucidate its molecular mechanisms and therapeutic potential in lung cancer [19].

## 5. ADCK3

AarF domain-containing protein kinase 3 (ADCK3), also known as coenzyme Q8A (COQ8A), is a protein-coding gene located on chromosome 10. The ADCK3 protein localizes to the mitochondrial cristae and is believed to play a critical role in maintaining mitochondrial homeostasis, as well as supporting various metabolic functions. Among the ADCK family members, ADCK3 has been the most extensively studied, particularly for its essential role in coenzyme Q10 (CoQ10) biosynthesis. CoQ10 plays a critical function in mitochondrial energy metabolism as it is a key component of the electron transport chain. Research has confirmed that ADCK3 possesses kinase activity with a biologically active kinase domain and is integral for CoQ10 synthesis. Mutations or loss of ADCK3 have been linked to neurological disorders and conditions associated with CoQ10 deficiency. Early studies indicate that CoQ10 supplementation may offer therapeutic benefits for patients with ADCK3-related deficiencies (N = 23). In particular, patients with autosomal recessive cerebellar ataxia 2 (ARCA2), a disorder caused by ADCK3 mutations, exhibited clinical improvement after a year-long CoQ10 supplementation regimen, suggesting its potential to moderately enhance motor function in ataxic individuals. However, further research is required to fully understand the therapeutic scope of CoQ10 supplementation in ADCK3-deficient patients [16,26,29,44,45,47,48,66,67,68].

A 2014 study highlights the transmembrane dimerization of the ADCK3 protein within *E. coli* membranes. It was noted that ADCK3 is abundant in smaller amino acids, such as glycine, alanine, and serine, which contribute to the structural integrity of the helices present in the membrane. The stability of ADCK3 dimers suggests potential kinase activity. In *S. cerevisiae*, ADCK3 has been characterized as an ATPase whose activity is enhanced by cardiolipin-rich membranes and phenolic intermediates of coenzyme Q (CoQ). This ATPase function is believed to support the CoQ biosynthetic complex by stabilizing its components or facilitating their proper assembly. This process likely occurs within the mitochondrial inner membrane, where ADCK3 plays a crucial role in maintaining the integrity and efficiency of CoQ biosynthesis [29,68].

In cell lines derived from autosomal recessive cerebellar ataxia type 2 (ARCA2) patients, ADCK3 has been observed to localize specifically to the mitochondrial cristae, reinforcing its critical role in CoQ biosynthesis and confirming its direct association with the CoQ biosynthetic complex. This localization aligns with ADCK3’s established function in mitochondrial energy metabolism, as CoQ serves as an essential electron carrier within the electron transport chain (ETC), facilitating ATP production. Disruptions in CoQ biosynthesis can severely impair oxidative phosphorylation, leading to mitochondrial dysfunction and subsequent cellular stress. In vitro, ADCK3 interacts with CoQ3, 5, 7, and 9. Mutations in ADCK3 resulted in compromised mitochondrial function, manifesting as increased oxidative stress, elevated lysosomal content, and a breakdown in mitochondrial homeostasis. These abnormalities suggest a failure in the cellular mechanisms responsible for maintaining mitochondrial quality control, leading to the accumulation of damaged organelles [26].

Additionally, ADCK3 mutant cells exhibit heightened sensitivity to hydrogen peroxide exposure, along with a significant increase in reactive oxygen species (ROS) and reactive nitrogen species (RNS) production. This indicates that ADCK3 plays a crucial role in mitochondrial redox homeostasis, potentially by regulating antioxidant defense mechanisms or mitochondrial quality control pathways. Excessive accumulation of reactive oxygen species (ROS) and reactive nitrogen species (RNS) can result in oxidative damage that affects mitochondrial DNA (mtDNA), proteins, and lipids. This damage further exacerbates mitochondrial dysfunction and contributes to neurodegeneration. Notably, a loss of ADCK3 is associated with increased oxygen consumption and glycolytic activity, which may serve as a compensatory response to mitochondrial stress. These findings underscore ADCK3’s crucial role in maintaining mitochondrial integrity, particularly through its essential involvement in coenzyme Q (CoQ) biosynthesis. Additionally, a study involving ARCA2 patients treated with CoQ10 supplementation (N = 4) over the course of a year demonstrated a mild improvement in motor activity. Understanding the molecular mechanisms underlying ADCK3 deficiency could pave the way for targeted therapeutic approaches, such as CoQ supplementation or mitochondrial-protective strategies, to mitigate the effects of mitochondrial dysfunction in ARCA2 and related disorders. Targeting ADCK3-related pathways may offer potential therapeutic strategies for mitochondrial dysfunction and associated disorders [26,30,47].

ADCK3 has been identified as a prognostic marker for hepatocellular carcinoma (HCC), with its expression levels correlating with tumor progression and the surrounding microenvironment. Patients with high ADCK3 expression exhibit increased tumor purity, indicating a greater proportion of cancerous cells within the tumor mass. This resulted from elevated immune scores, stromal scores, and ESTIMATE scores, the latter being a combined metric of immune and stromal scores that indicate infiltration of stromal and immune cells. These findings suggest that ADCK3 may influence the composition of the tumor microenvironment by modulating immune cell infiltration, stromal interactions, and extracellular matrix remodeling, all of which are critical factors in tumor growth, immune evasion, and therapeutic response. Additionally, ADCK3 has been correlated with the PI3K/Akt signaling pathway, a key regulator of tumorigenesis and cancer progression [49].

Beyond HCC, studies in endometrial carcinoma (EC) have also highlighted ADCK3 as a potential biomarker. Notably, its expression has been linked to patient response to medroxyprogesterone acetate (MPA) therapy, with evidence indicating that ADCK3 acts as a key regulator of MPA-induced cell death. ADCK3 expression has been observed to play a role in MPA-induced ferroptosis through the upregulation of ALOX15. This suggests that ALOX15 may serve as a downstream effector of ADCK3 in this process. Additionally, in EC cells, ADCK3 appears to be transcriptionally regulated by p53, indicating that ADCK3 is a direct downstream target of p53. These findings suggest that ADCK3 may serve as both a biomarker and a therapeutic target in human cancers. However, further mechanistic studies are required to validate its role in tumorigenesis across different cancer types [46].

## 6. ADCK4

AarF domain-containing protein kinase 4 (ADCK4), also known as coenzyme Q protein 8B (COQ8B), is a protein-coding gene located on chromosome 19 at position q13.2 (19q13.2). The ADCK4 protein localizes to the inner mitochondrial membrane, specifically within the cristae, where it is believed to play a role in mitochondrial metabolism. Similar to ADCK3, ADCK4 has been extensively studied for its role in CoQ biosynthesis and mitochondrial function. In *E. coli*, ADCK4 exhibits 50% sequence identity with ADCK3, and their transmembrane domains are nearly identical. The amino acid compositions of the two proteins are also similar, suggesting that ADCK4 may play a role in maintaining the structural integrity of membranes. It is proposed that there is compatibility between ADCK3 and ADCK4, potentially facilitating the formation of a heterodimeric complex between these two proteins [16,17,27,29,50,69,70].

Mutations in ADCK4 have been identified as a genetic contribution to steroid-resistant nephrotic syndrome (SRNS), a progressive kidney disorder characterized by proteinuria, podocyte dysfunction, and resistance to corticosteroid treatment. These mutations are also associated with CoQ10 deficiency, which further exacerbates renal pathology by impairing mitochondrial energy metabolism and antioxidant defenses. In in vivo studies, the loss of ADCK4 led to reduced survival rates and the development of severe focal segmental glomerulosclerosis (FSGS), a condition marked by glomerular scarring, podocyte damage, and progressive kidney failure [16,17,27,29,50,69,70].

The knockout of ADCK4 in animal models significantly decreased CoQ10 levels, resulting in mitochondrial dysfunction, impaired oxidative phosphorylation, and increased oxidative stress, findings that are consistent with the roles of other ADCK isoforms in maintaining mitochondrial homeostasis. Treatment with 2,4-dihydroxybenzoic acid (2,4-diHB) was able to partially rescue the mitochondrial defects caused by ADCK4 deficiency, suggesting a potential therapeutic strategy for mitochondrial-associated kidney diseases. These findings highlight the essential role of ADCK4 in maintaining renal function, mitochondrial integrity, and CoQ biosynthesis, emphasizing the need for further research to explore targeted interventions for patients with ADCK4-related nephropathy and CoQ deficiencies [30,53,54].

The link between ADCK4 and CoQ10 is reinforced by studies examining ADCK4-deficient diseases and the potential of CoQ10 as both a biomarker and a therapeutic intervention. In glomerulopathy patients, urinary CoQ10 levels have been explored as a biomarker to assess disease progression and predict clinical remission. Patients with elevated urinary CoQ10-to-creatinine ratios may have underlying CoQ10 deficiencies, suggesting that targeted CoQ10 supplementation could improve their clinical outcomes by restoring mitochondrial function and mitigating oxidative stress. Additionally, patients with SRNS or chronic kidney disease (CKD) have demonstrated positive responses to CoQ10 therapy, with improvements in renal function evidenced by reduced proteinuria and an increased estimated glomerular filtration rate (eGFR). These findings underscore the renal-protective effects of CoQ10, which have been sustained even in long-term follow-up studies, highlighting its potential as a therapeutic strategy for ADCK4-related nephropathies [30,53,54].

The role that ADCK4 plays in renal and mitochondrial dysfunction is further supported by findings from a 2020 study, which investigated its interactions with mitochondrial proteins and its impact on renal function. ADCK4 was found to interact primarily with coenzyme Q5 (CoQ5), an enzyme essential for ubiquinone CoQ biosynthesis, reinforcing its role in mitochondrial homeostasis. The loss of ADCK4 in podocytes resulted in destabilization of the CoQ biosynthetic complex, leading to mitochondrial dysfunction, cytoskeletal disorganization, and impaired podocyte integrity. These disruptions contribute to the progression of glomerular disease and the development of proteinuria- hallmarks of SRNS [50].

The ability of ADCK4 to stabilize the coenzyme Q (CoQ) biosynthetic pathways is primarily attributed to its regulatory role in protein–protein interactions. Experimental evidence suggests that the depletion of ADCK4 results in a significant reduction of oxidative phosphorylation complex II–II proteins, subsequently impairing mitochondrial respiration. Additionally, the interaction between ADCK4 and CoQ5 is critical for the stabilization of the CoQ complex as well as the maintenance of the cytoskeletal structure. Importantly, treatment with 2,4-diHB in this study mitigated these defects, promoting podocyte survival, restoring mitochondrial function, and improving overall renal function. This is similar to previous data on 2,4-diHB treatment. These findings highlight the potential of 2,4-diHB as a therapeutic agent for ADCK4-deficient nephropathies, offering a promising approach to stabilizing podocytes and preserving kidney function in patients with SRNS. Further research is needed to establish and elucidate the precise molecular mechanisms by which CoQ10 supports mitochondrial integrity and podocyte function in ADCK4-deficient kidney disorders. Nonetheless, these findings reinforce the essential role of ADCK4 in renal and mitochondrial functions [50].

The role of ADCK4 in human cancers remains largely unexplored, with limited studies investigating its potential involvement in tumorigenesis. A 2019 study identified a fusion between ADCK4 and NUMB-like endocytic adaptor protein (NUMBL) in cutaneous squamous cell carcinoma (cSCC), suggesting a potential oncogenic or tumor-modulating role. Data from this study indicated that ADCK4- and NUMBL-positive cells were smaller in size and exhibited a slower disease progression compared to ADCK4- and NUMBL-negative groups. The knockdown of ADCK4 and NUMBL expression resulted in a mild reduction in cellular proliferation, implying a possible role in tumor growth regulation. Additionally, the ADCK4–NUMBL complex was also identified in sinonasal tract mucosal melanoma (SNTMM); however, its biological function and impact on tumor progression remain uncharacterized. These findings suggest that ADCK4 may have context-dependent roles in cancer, but further research is required to elucidate its molecular mechanisms, potential oncogenic functions, and therapeutic relevance across various malignancies [51,52].

## 7. ADCK5

AarF domain-containing kinase 5 (ADCK5) is a protein-coding gene located on chromosome 8 at position q24.3 (8q24.3). Similarly to other ADCK isoforms, the ADCK5 protein localizes to the mitochondria, where it is implicated in the regulation of mitochondrial homeostasis, potentially contributing to the maintenance of mitochondrial function and homeostasis. A 2024 study identified a strong interaction between ADCK5 and mitochondria-related genes (MRGs) in childhood allergic asthma, suggesting a potential link between mitochondrial dysfunction and immune dysregulation in the disease. These interactions may contribute to impaired mitochondrial function, which has been associated with inflammation, oxidative stress, and altered metabolic signaling in asthma pathophysiology. Further research is needed to establish a causal relationship between ADCK5 expression and asthma pathophysiology [16,28,71].

Several studies have identified a potential link between ADCK5 expression and the tumorigenesis of various malignancies. In lung cancer, ADCK5 overexpression has been associated with increased invasion and migration, suggesting a role in tumor progression and metastasis. This effect may be mediated through the interaction between ADCK5 and pituitary tumor-transforming gene-1 (PTTG1), a well-established oncogene involved in cell cycle regulation and tumor development. Although ADCK5 does not appear to act directly on PTTG1, emerging evidence suggests that ADCK5 may regulate the transcription of PTTG1, potentially influencing its downstream oncogenic effects. Furthermore, ADCK5 has been implicated in the phosphorylation of SOX9 at Ser181, a modification known to be critical for tumor cell proliferation, differentiation, and survival. These findings imply that ADCK5 plays a pro-oncogenic role in lung cancer progression and provides a possible mechanistic casual for its oncogenic activity [22].

A study on pancreatic cancer suggests that ADCK5 may play a role in modulating CD73-related pathways; however, the precise mechanisms underlying this interaction remain unclear [55].

Early studies investigating the role of ADCK5 in drug resistance suggest that it may be a promising target for combination therapy in patients who have developed resistance to JQ1, a bromodomain inhibitor. Research indicates that JQ1 treatment leads to the overexpression of several kinases, including ADCK5, potentially contributing to therapy resistance. Targeting ADCK5 in conjunction with JQ1 therapy may enhance treatment efficacy and help overcome resistance mechanisms [56].

Other studies have identified ADCK5 as a potential biomarker for disease prognosis. A 2023 study discovered a prognostic signature consisting of eight senescence-related genes, including ADCK5, suggesting its involvement in tumor progression and cellular senescence. ADCK5 was found to be upregulated in prostate tumor tissues and implicated in the induction of senescence, a process that can have both tumor-suppressive and tumor-promoting effects depending on the cellular context. The identification of ADCK5 as a prognostic marker highlights its potential clinical utility in predicting patient sensitivity to immunotherapies and drug susceptibilities. Understanding its role in tumor senescence could provide valuable insights into personalized treatment strategies, making it a promising target for further research in cancer prognosis and therapeutic response [57].

## 8. Therapeutic Potential of ADCK Kinases

As of this writing, ADCK3 (also known as COQ8A) remains the only isoform within the ADCK family for which small-molecule inhibitors have been successfully identified and characterized. Recent studies using structure-based virtual screening have led to the discovery of several potent and selective ADCK3 inhibitors, highlighting the druggability of this atypical kinase. These inhibitors not only demonstrate biochemical activity and high selectivity over other kinases such as p38, but also exhibit stable binding within the ADCK3 active site, as validated using molecular dynamics and metadynamics simulations. This progress underscores the therapeutic potential of targeting ADCK3, particularly in disorders involving CoQ10 deficiency, mitochondrial dysfunction, or ADCK3-associated ataxia. Furthermore, the development of chemical probes against ADCK3 sets a precedent for exploring the therapeutic relevance of other ADCK isoforms, encouraging further drug discovery efforts across the family [72,73,74].

In contrast, the other ADCK isoforms remain structurally uncharacterized, which significantly limits structure-based drug design approaches. Moreover, these isoforms are poorly understood at the functional level, with incomplete information on their enzymatic activity, physiological substrates, and disease-specific roles. This lack of functional annotation not only impedes the development of robust biochemical or cellular assays but also reduces pharmaceutical interest due to the unclear therapeutic relevance. Without clear biological targets or disease associations, these kinases are often deprioritized in drug discovery pipelines. As a result, the perceived druggability of ADCK1, ADCK2, ADCK4, and ADCK5 remains low, highlighting a critical gap in our understanding of this protein family and underscoring the need for deeper structural and functional investigation. Although less characterized than ADCK3, other ADCK are increasingly linked to critical processes including mitochondrial dynamics, ROS regulation, immune modulation, and cancer progression, suggesting they may represent promising, yet underexplored, targets for therapeutic intervention. Continued functional and structural studies could reveal isoform-specific vulnerabilities that can be leveraged for drug development in diseases marked by mitochondrial dysfunction and metabolic imbalance.

## 9. Conclusions

The regulation of cellular bioenergetics is crucial in several metabolic diseases, particularly many types of human cancers. Energy production pathways, such as glycolysis and oxidative phosphorylation, are often dysregulated in cancer. Additionally, mitochondrial signaling and reactive oxygen species (ROS) management play vital roles in the regulation of tumor growth and drug resistance [75]. The ADCK (AarF domain-containing kinase) protein family plays a multifaceted role in maintaining mitochondrial integrity and regulating cellular metabolism, with each member exhibiting both unique and overlapping functions. As illustrated in Figure and summarized in the accompanying Table 1, ADCK1–ADCK5 are involved in processes such as CoQ biosynthesis, mitochondrial dynamics, oxidative stress response, immune regulation, and apoptosis. These kinases directly or indirectly influence critical signaling pathways, including PI3K/Akt, mTOR, and NF-κB. The dysregulation of ADCK proteins is implicated in a broad spectrum of diseases, ranging from rare inherited disorders such as cerebellar ataxia and nephrotic syndromes to common malignancies including lung, breast, colon, and prostate cancers. The oncogenic roles of these kinases extend to promoting cell proliferation, metastasis, drug resistance, and immune evasion, often via their impact on mitochondrial bioenergetics and redox balance.

Across all ADCK isoforms, alterations in their expression or function have been shown to disrupt mitochondrial homeostasis, though through distinct and diverse pathways. These include regulation of mitochondrial fusion and fission dynamics (particularly by ADCK1), modulation of lipid homeostasis (as seen with ADCK1 and ADCK2), and lysosomal remodeling (notably involving ADCK3 and ADCK4). Despite their unique mechanisms, a unifying feature among all five ADCK proteins is their influence on reactive oxygen species (ROS) production. Elevated ROS levels, often a consequence of mitochondrial dysfunction, can lead to oxidative damage of DNA, proteins, and lipids—processes that are central to the initiation and progression of various diseases, including cancer. Furthermore, each ADCK isoform has been implicated in at least one energy-related disease or cancer-related process, ranging from tumor modulation and metastasis to immune evasion and drug resistance. This collective involvement underscores the broader significance of the ADCK family in regulating mitochondrial function, cellular metabolism, and disease pathogenesis.

Among the ADCK family members, ADCK2, ADCK3, and ADCK5 are notably involved in modulating immune system functions, influencing key processes such as the inflammatory response, immune regulation, and immune evasion. These immunomodulatory roles suggest that certain ADCK isoforms contribute to shaping the tumor microenvironment and may affect susceptibility to immune-based therapies. In parallel, ADCK2, ADCK3, and ADCK4 are directly implicated in the biosynthesis of coenzyme Q (CoQ), an essential lipid-soluble component of the mitochondrial electron transport chain. CoQ plays a vital role in oxidative phosphorylation and ATP production, linking these isoforms to the regulation of mitochondrial energy metabolism and redox balance.

Taken together, the ADCK protein family exhibits significant functional convergence in maintaining mitochondrial integrity, controlling ROS levels, and supporting critical cellular processes including energy production, immune responses, and oncogenic transformation. This functional overlap highlights the ADCK family not only as potential biomarkers but also as a promising set of targets for therapeutic intervention in a wide range of diseases characterized by mitochondrial dysfunction, metabolic dysregulation, and immune escape mechanisms, such as cancer and metabolic syndromes. Despite recent progress, many aspects of ADCK functions, regulation, and interactions remain poorly understood. Future research aimed at elucidating the structural and regulatory mechanisms of ADCKs will be crucial for developing targeted interventions in both oncology and metabolic medicine.

## Figures and Tables

**Figure 1 ijms-26-05783-f001:**
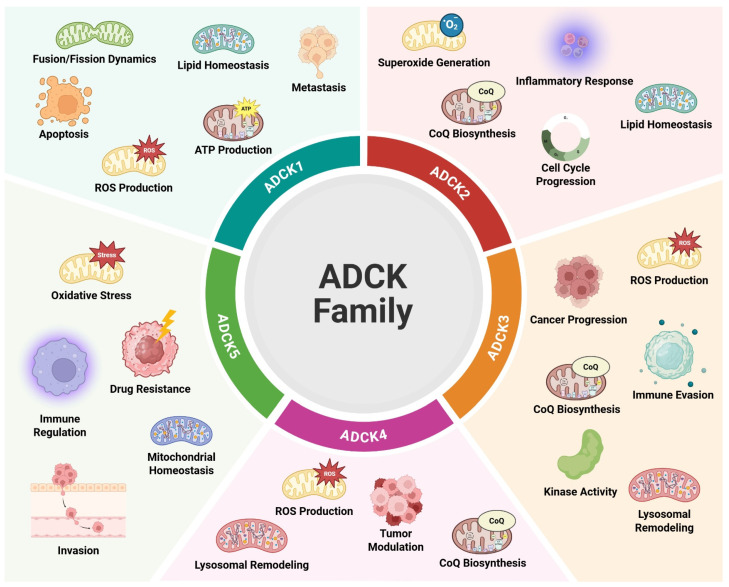
ADCK family functions. Distinct yet overlapping roles of ADCK family members (ADCK1–ADCK5) in cellular homeostasis and tumorigenesis. The ADCK kinases are involved in a variety of mitochondrial functions, including coenzyme Q (CoQ) biosynthesis, mitochondrial dynamics, and energy production. ADCK1 is implicated in mitochondrial fusion/fission dynamics, lipid homeostasis, apoptosis, ATP production, ROS production, and metastasis. ADCK2 contributes to CoQ biosynthesis, lipid regulation, superoxide generation, inflammatory signaling, and cell cycle progression. ADCK3 is associated with CoQ biosynthesis, ROS production, immune evasion, and cancer progression. ADCK4 participates in tumor modulation, CoQ biosynthesis, lysosomal remodeling, and ROS regulation and exhibits kinase activity. ADCK5 is linked to oxidative stress response, mitochondrial homeostasis, immune regulation, drug resistance, and cellular invasion. Collectively, the ADCK family modulates key processes underlying oncogenesis, metabolic adaptation, immune signaling, and mitochondrial integrity, highlighting their potential as diagnostic markers and therapeutic targets in a range of pathological contexts. Created in BioRender. Jacquet, N. (2025) https://BioRender.com/bt89fbt (accessed on 8 June 2025).

**Table 1 ijms-26-05783-t001:** Summary of the functional role, localization, and pathological implications of ADCK kinases.

Protein	Functions	Proposed Targets/Interactions	Localization	Pathological Implications	Potential Oncogenic Role
ADCK1	Mitochondrial Maintenance/ Homeostasis [18,36]	Directly- COQ3-4-5-6-9, PDSS2, MT-CO1, NDFUS3, NDFUV2, YME1L1 Indirectly-OPA1, IMMT, TCF4, B-Catenin [18,25,33,36]	Mitochondrial Membrane [36]	Colon Cancer, Osteosarcoma, Parathyroid Cancer, Schizophrenia [33,36,37,38]	Tumor Growth/Cell Proliferation, ATP Production, Cellular Migration, Apoptosis, Colony Formation, Metastasis [33,38]
ADCK2	CoQ Biosynthesis, Mitochondrial Metabolism/Maintenance [23,39]	Directly- COQ3-4-5-9, Akt Indirectly- MYL6/TNFa-HIF-1a axis/RELB-dependent NF-KB signaling pathway, Akt-mTOR signaling pathway, S6K [17,19,23,34,35,40]	Mitochondrial Matrix [41]	Breast Cancer, CoQ Deficiency, Melanoma, Osteo- sarcoma, Non- Small Cell Lung Cancer [34,35,40,42,43]	Tumor Progression, Oxidative Stress, ROS Production, Inflammatory Response, Tumor Growth/Cell Proliferation, Metabolic Adaptation, Cellular Motility, Invasion, Cell Cycle Progression, Cell Survival [32,34,35,40]
ADCK3	CoQ Biosynthesis, Kinase Activity, Mitochondrial Maintenance/ Homeostasis [26,29]	Directly- COQ2-3-4-5-6-7-9- 10, PDSS1, PDSS2, MBP, p53, ALOX15, UNC-CA157, IMM Indirectly-PI3K/Akt, MPA, OX- PHOS [20,24,26,44,45,46]	Mitochondrial Cristae, Inner Mitochondrial Membrane/Cristae [26,29]	Autosomal Recessive Cerebellar Ataxia 2, Chronic Kidney Disease, Endometrial Carcinoma, Hepatocellular Carcinoma, CoQ Dysfunction [44,45,46,47,48,49]	Tumor Purity, Tumor Progression, Tumor Growth/Cell Proliferation, Immune Evasion, Drug Resistance, MPA-Induced Cell Death [46,49]
ADCK4	CoQ Biosynthesis, Mitochondrial Metabolism/ Maintenance [27,29]	Directly- COQ2-3,4,5,6,7,9,10A, PDSS1, PDSS2, NUMBL Indirectly- BRAF, RAS [21,27,50,51,52]	Inner Mitochondrial Membrane/Cristae Mitochondria and Cytosol [21,29]	Focal Segmental Glomerulosclerosis, Nephropathy, Sinonasal Tract Mucosal Melanoma, Steroid-Resistant Nephrotic Syndrome CoQ Dysfunction [31,50,51,52,53,54]	Tumor Modulation, Tumor Growth/Cell Proliferation [51,52]
ADCK5	Mitochondrial Maintenance/ Homeostasis [28]	Directly- SOX9, SCLC52A2, MFN1, CD73 Indirectly- PTTG, NBR1, BNIP3, STX17 [22,28,55,56]	Mitochondria [16]	Asthma, Lung Cancer, Pancreatic Cancer, Prostate Cancer [55,56,57]	Tumor Growth/Cell Proliferation, Tumor Progression, Metastasis, Cell Cycle Regulation, Cellular Differentiation, Cell Survival, Drug Resistance [55,56,57]
References: [16,17,18,19,20,21,22,23,24,25,26,27,28,29,30,31,32,33,34,35,36,37,38,39,40,41,42,43,44,45,46,47,48,49,50,51,52,53,54,55,56,57]					

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
