# Peer review of "The ADCK Kinase Family: Key Regulators of Bioenergetics and Mitochondrial Function and Their Implications in Human Cancers"

_ijms, 2025, doi:10.3390/ijms26125783_

Round 1
Reviewer 1 Report
Comments and Suggestions for Authors
This review article offers a comprehensive examination of the AarF domain-containing kinase family, with a particular emphasis on their roles in mitochondrial bioenergetics and their emerging implications in human cancers. The topic is timely and relevant, addressing a growing area of interest in mitochondrial biology and oncology. The manuscript is generally well-organized, progressing logically from an overview of mitochondrial function to detailed discussions of each ADCK isoform. The conclusion effectively summarizes the central findings and underscores the importance of continued research in this area. By highlighting the therapeutic potential of targeting ADCK kinases in the context of cancer metabolism, this review contributes meaningfully to the field.
Despite its strengths, the manuscript requires slight refinement on certain areas listed below. Several critical and technical issues must be addressed to improve the manuscript's rigor, coherence, and overall impact. Notably, the review lacks structural synthesis across the ADCK isoforms, which could facilitate clear distinction between evidence-based conclusions and speculative interpretations.
Here are my detailed comments and suggestions:
- Many sections are verbose and reiterate general mitochondrial biology (e.g., ATP production, oxidative phosphorylation) without sufficiently anchoring the discussion to ADCK-specific insights. This dilutes the focus and weakens the scientific narrative.
- Consider condensing the introductory and background sections to remove redundancy. Prioritize mechanistic insights and highlight how individual ADCK isoforms differ or overlap in function, localization, and disease relevance.
- Each ADCK isoform is treated in isolation, with minimal cross-referencing or comparative evaluation. This limits the reader’s ability to understand the broader functional landscape of the ADCK family.
- Include a synthesis or summary section that compares the known functions, molecular pathways, and disease links of ADCK1–ADCK5 for visualizing similarities and differences across isoforms.
- It would be interesting to expand the discussion on therapeutic opportunities: How might ADCK kinases be targeted pharmacologically? What are the limitations of current understanding? Are any of these kinases druggable?
- The discussion on therapeutic opportunities is underdeveloped. Consider addressing the following questions: a. Are any ADCK kinases considered druggable? b. What therapeutic strategies or small molecules have been explored? c. What are the current limitations in targeting these kinases in clinical contexts?
- The discussion on therapeutic opportunities is underdeveloped. Consider addressing the following questions as feasible: a. A schematic overview of ADCK kinase localization and function within mitochondria. b. A disease-association map linking specific ADCK isoforms to cancers and other disorders. c. A diagram of the CoQ biosynthesis pathway highlighting ADCK involvement.
Author Response
Thank you for your valuable comments, please see the attachment.

Reviewer 2 Report
Comments and Suggestions for Authors
The review promises an authoritative synthesis yet fundamental issues undermine its value. There is no description of how the literature was gathered, screened, or prioritised, so readers cannot judge completeness.
A table and a figure that the text explicitly relies on are missing (Table 1 and Figure 1, lines 84-102) Much of the opening section re-explains standard bioenergetics instead of analysing ADCK-specific mechanisms; key structural work on the atypical kinase fold of COQ8A/B is cited but never discussed.
Case associations are reported as though causal—polymorphisms in schizophrenia and asthma, or expression correlations in breast cancer—without evaluating study size, replication, or confounders. The conclusion then claims these data “pave the way for targeted therapies” without addressing druggability or past failures (lines 421-429).
Many citations are GeneCards or NCBI pages rather than peer-reviewed studies, numbering drifts after ref 60, and several entries repeat. Important recent primary papers are absent. Stylistically, long sentences, repeated paragraphs, inconsistent acronyms (CoQ/Coenzyme Q10) and mis-matched section numbering impede readability; the header still labels the piece an “Article” although it is a review. Biological accuracy also slips—for example, the blanket statement that all isoforms have confirmed kinase activity, or that ADCK4 mutations are “the” cause of steroid-resistant nephrotic syndrome, ignoring other podocyte genes.
To reach publishable quality the authors should: provide a transparent search strategy; supply the missing visuals and redesign them to show domain architecture, interactomes, and disease variants; replace database URLs with peer-reviewed sources and correct reference order; trim textbook material and add mechanistic depth, especially structural and regulatory insights; temper speculative therapeutics; standardise terminology; and undertake professional language editing. Addressing these points will transform the manuscript from a catalogue into a critical, reliable resource.
Author Response

(The authors gave the same response as above.)

Reviewer 3 Report
Comments and Suggestions for Authors
The author’s purpose of the review article about “The ADCK Kinase Family: Key Regulators of Bioenergetics and 3 Mitochondrial Function, and Their Implications in Human Cancers” is interesting also from related research fields. The paper is globally easy to follow and understand. Suggestions:
- References 4 and 5 although well-known books, should be replaced by recent papers in the field
- Table 1 is to small. Should be vertical and all the page. ]he references used at the footnotes of the table should be not as it is….Acosta et al 2019 it should be [15.]
- Localization parameter in Table I is not clear. Mitochondrial membrane means inner or outer? The localization just mitochondria means what? Please clarify.
- Figure 1 is not clearly described and/or explained in the body text.
- Figure 1 has no legend. Please clarify.
- The symbols at Figure should be place in the right side: 1) ROS; 2) cell cycle; 3) cancer progression….up to 17?.
- For The symbols at Figure 1, mitochondria symbol for superoxide, ROS production; snf oxidative stress is not the same?
- Figure 1 is for effects observed or for mechanisms and/or modes of action?
- At the end of page 6 the authors referred to Akt-mTOR signalling but no symbol for this is at Figure 1? Please clarify.
- Before conclusion I would like to suggest a timeline figure within the major events described above….first studies….
- At conclusion section the authors refer to the involvement of ADCK proteins in : 1) mitochondrial; coenzyme Q biosynthesis and 3) diseases. But where is the figure reflecting and summaryzing this conclusion? Please clarify.
- The last 3 sentences of the conclusions are not specific and could be apply for any conclusion by just changing the key word. Could it be more specific?
- Conclusions section could be eventually organized in order to highlited the research described with partial conclusions in the order of the paper presentation and than with global conclusions and future perspectives.
Author Response
Reviewer 2:
The author’s purpose of the review article about “The ADCK Kinase Family: Key Regulators of Bioenergetics and 3 Mitochondrial Function, and Their Implications in Human Cancers” is interesting also from related research fields. The paper is globally easy to follow and understand. Suggestions:
- References 4 and 5 although well-known books, should be replaced by recent papers in the field
Response: Textbook material has been trimmed.
- Table 1 is to small. Should be vertical and all the page. ]he references used at the footnotes of the table should be not as it is….Acosta et al 2019 it should be [15.]
Response: Table 1 has been resized, and the reference section has been revised.
- Localization parameter in Table I is not clear. Mitochondrial membrane means inner or outer? The localization just mitochondria means what? Please clarify.
Response: Localization is specified to the level of detail available in the current literature. localization to mitochondria means the proteins have been identified as mitochondrial proteins, but not with specific localization to a particular compartment of the mitochondria.
- Figure 1 is not clearly described and/or explained in the body text.
Response: We have included a more in-depth text to explain the figure.
- Figure 1 has no legend. Please clarify.
Response: We had a short figure legend. But we have improved the legend with more detailed information.
- The symbols at Figure should be place in the right side: 1) ROS; 2) cell cycle; 3) cancer progression….up to 17?.
Response: We have made the changes accordingly.
- For The symbols at Figure 1, mitochondria symbol for superoxide, ROS production; snf oxidative stress is not the same?
Response: Different symbols were intentionally used to reflect the specific terminology and mechanistic distinctions reported in the original source studies, and thus are not meant to be interchangeable. For clarification, "oxidative stress" denotes the general imbalance between the production and accumulation of reactive oxygen species (ROS), while the ROS symbol in the figure specifically represents direct involvement in ROS generation. "Superoxide" is a specific subset of ROS; although multiple ROS species exist, only superoxide production was implicated in relation to ADCK1 in the cited study. Therefore, the terminology and associated symbols were used with precision to preserve the specificity of the reported findings.
- Figure 1 is for effects observed or for mechanisms and/or modes of action?
Response: Figure 1 is intended to represent both observed effects and proposed mechanisms or modes of action, as reported in the primary literature. Each symbol or label within the figure reflects either a direct functional outcome (e.g., cell proliferation, ROS production) or a mechanistic pathway (e.g., mitochondrial regulation, kinase activity) attributed to individual ADCK family members.
- At the end of page 6 the authors referred to Akt-mTOR signalling but no symbol for this is at Figure 1? Please clarify.
Response: Symbols for cellular processes regulated by these pathways, such as cell cycle progression and proliferation, are included in the chart- no specific signaling pathway is listed in the figure for any of the proteins.
- Before conclusion, I would like to suggest a timeline figure within the major events described above….first studies…
Response: Thank you for the suggestion. The ADCK family is relatively new and most of the studies were completed recently; that’s why we did not include a timeline in the manuscript.
- At conclusion section the authors refer to the involvement of ADCK proteins in : 1) mitochondrial; coenzyme Q biosynthesis and 3) diseases. But where is the figure reflecting and summaryzing this conclusion? Please clarify.
Table 1 lists the role of ADCK proteins in mitochondrial functions, coenzyme Q biosynthesis, and implicated diseases.
- The last 3 sentences of the conclusions are not specific and could be apply for any conclusion by just changing the key word. Could it be more specific?
Response: The paper’s conclusion has been improved to be more comprehensive.
- Conclusions section could be eventually organized in order to highlited the research described with partial conclusions in the order of the paper presentation and than with global conclusions and future perspectives.
Response: The paper’s conclusion now follows the order of the sub-sections of the manuscript: mitochondrial functions, disease implications, and role in human cancers.
Round 2
Reviewer 2 Report
Comments and Suggestions for Authors
All major concerns raised have been addressed satisfactorily in the revised manuscript. The authors made significant improvements to the transparency, structure, scientific depth, and clarity of their review article.
Reviewer 3 Report
Comments and Suggestions for Authors
No further suggestions. it is not clear in the new version where were the new changes done.